# Information Complexity Ranking: A New Method of Ranking Images by Algorithmic Complexity

**DOI:** 10.3390/e25030439

**Published:** 2023-03-01

**Authors:** Thomas Chambon, Jean-Loup Guillaume, Jeanne Lallement

**Affiliations:** 1Laboratoire Informatique, Image et Interaction (L3i), La Rochelle University, 23 Avenue Albert Einstein, 17000 La Rochelle, France; 2Laboratoire Usages du Numerique Pour le Developpement Durable (NUDD), La Rochelle University, 39 rue de Vaux De Foletier, 17000 La Rochelle, France

**Keywords:** algorithmic information theory, information complexity, similarity complexity, Kolmogorov complexity

## Abstract

Predicting how an individual will perceive the visual complexity of a piece of information is still a relatively unexplored domain, although it can be useful in many contexts such as for the design of human–computer interfaces. We propose here a new method, called Information Complexity Ranking (ICR) to rank objects from the simplest to the most complex. It takes into account both their intrinsic complexity (in the algorithmic sense) with the Kolmogorov complexity and their similarity to other objects using the work of Cilibrasi and Vitanyi on the normalized compression distance (NCD). We first validated the properties of our ranking method on a reference experiment composed of 7200 randomly generated images divided into 3 types of pictorial elements (text, digits, and colored dots). In the second step, we tested our complexity calculation on a reference dataset composed of 1400 images divided into 7 categories. We compared our results to the ground-truth values of five state-of-the-art complexity algorithms. The results show that our method achieved the best performance for some categories and outperformed the majority of the state-of-the-art algorithms for other categories. For images with many semantic elements, our method was not as efficient as some of the state-of-the-art algorithms.

## 1. Introduction

Information complexity estimation is used in many contexts to classify texts [1], reconstruct phylogenetic trees [2], or analyze the behavior of financial markets [3]. Information complexity is also used in the field of image analysis for image classification or comparison [4]. Information complexity estimation can also be applied to gain a better understanding of how individuals perceive complexity and how they respond to different levels of complexity [5,6].

To evaluate this impact, it is first important to quantify and qualify the content of the information produced. However, the prerequisite for solving this problem is in the definition of the term “information”.

To quantify the information content, two methods are generally used, the Shannon entropy (1948) and Kolmogorov complexity (1965). C.E. Shannon was the first to define information based on the notion of entropy. However, this method’s main limitation is its ability to evaluate the complexity of an image. Shannon’s observational point of view does not allow a comparison of multiple images because the entropy is derived from a summation of the components of the histogram, which represent the frequencies of appearance of the symbols composing the message and thus does not take into account the order of these symbols (i.e., the content of the message).

A.N. Kolmogorov, inspired by Shannon’s theory, tried to provide another definition of the complexity of an object. The notion of complexity according to Kolmogorov is based on the algorithmic notions of the program and universal language. This approach can be summarized as follows: what is simple, can be briefly described. This new point of view characterizes the randomness of information, but in image analysis, the notion of structure is important and is not fully taken into account. To overcome these limitations, the notion of similarity between images can help to highlight a possible structure.

To calculate the similarity between images, the normalized compression distance (NCD) [1] is appropriate in the algorithmic context. However, NCD alone considers the distance between images but does not allow for the ranking of these images from the least to the most complex. To overcome this limitation, we propose a new method for ranking images according to their complexity that integrates the notion of similarity with NCD and the Kolmogorov complexity. The aim of our proposed method is to measure the complexity of each image and compare them to establish a ranking.


**Contributions:**
We propose a new method for classifying images according to their complexity known as ICR.We define a protocol for generating synthetic images in order to evaluate the performance of the compression algorithms that can be used with our new method.We perform experiments on a dataset of 1400 diverse images to prove the robustness of our approach.


The theoretical aspects of this method and their extension to the concept of information distance will be developed in Section 2. We present the new ranking method and formally prove that it is a partially ordered set in Section 3. Two experiments are proposed to prove the robustness of our proposal in Section 4. The conclusions of our work are summarized in Section 5.

## 2. Materials and Methods

### 2.1. State-of-the-Art Algorithms

#### 2.1.1. Statistical View of Complexity

One of the first efforts at qualifying the information was Shannon’s information theory. In his paper “A mathematical theory of communication” (1948), Shannon tackles the issue of message communication between a sender and a recipient [7], taking inspiration from probability theory, statistics, and thermodynamics. He introduces the concept of entropy in the field of information theory, where the idea is to measure the uncertainty of a transmitted message. The less frequent a message, the greater the amount of storage required to transmit it. Entropy is then interpreted as a measure of the number of bits required to encode the information to be sent and is defined as:(1)$$H\left(m\right)=n\left(-\sum p_{i}\log_{2}p_{i}\right)$$
where *m* is the message to transmit, *n* is the number of symbols in the message, and $p_{i}$ ($p_{i}=k_{i}/k$) is the relative frequency of symbol *i* in *m* (*k* is the number of symbols in the alphabet, i.e., (a-z) and $k_{i}$ is the number of occurrences of symbol *i* in *m*).

This formulation only takes into account the relative frequencies of the different symbols and the entropy is therefore not interested in the meaning of the transmitted message or, to cite Shannon himself, “These semantic aspects of communication are irrelevant to the engineering problem.” [7]. The frequencies also do not determine the order of symbols in a text or their 2D position in an image (the symbols of an image are the values of each pixel), whereas the complexity of a text or an image depends on this order or these positions [8].

In practice, the information used in our case (image classification) is not diverse enough for the statistical method to be interesting. It is then useful to focus on the notion of description with the following postulation: the more complex an object (i.e., rich in information), the longer its description. The unambiguous description of an object thus makes it possible to quantify its information content and, consequently, its complexity.

#### 2.1.2. Algorithmic View of Complexity

The evolution of this conceptual idea toward formal mathematical definitions has allowed the development of the concept of “Algorithmic Information Theory” (AIT) [9]. This concept was initiated by several works of Chaitin or Solomonoff, but mainly by A.N. Kolmogorov in the seminal paper “Three Approaches to Information” (1965) [10]. AIT is concerned with measuring string complexity but can also measure any other data structures. This algorithmic theory is inspired by probability theory, information theory, and randomness notions [11]. To implement this approach to information, Kolmogorov was concerned with the way the information is generated. As a result of his contribution, this new form of evaluating the information is generally named the “Kolmogorov complexity”.

To generate information, an algorithm can be used and executed in any programming language. Kolmogorov’s complexity calculation involves considering the size of the program and the algorithm that generates the information. The length of a program (description) allows us to measure the object’s complexity and, therefore, the content of the information.

The Kolmogorov complexity [10] of an object *x*, namely Kφ(x), is thus defined as the length of the shortest program that can compute *x*. This program can be written in any universal description language (Turing-complete programming languages such as C, C++, Python, JAVA).
(2)Kφ(x)=min{|p|:φ(p)=x}
where *p* is a computer program (with no input), φ executes programs (i.e., φ is a programming language plus a compiler plus the machinery to run programs), and φ(p) is the output of the execution of program *p*. Thus, for x∈O,Kφ(x) is the length of the shortest program p with which φ computes *x* [12].

According to Turing’s work on the undecidability of program termination [13] or Solmonoff’s later paper “A formal theory of inductive inference” [14], Kφ(x) is not effectively computable in the sense of Turing, because it is not recursive [15]. Nevertheless, Kφ(x) is approachable. In particular, in his invariance theorem [10], Kolmogorov showed that given any couple of programs (U,V) implementing the same algorithm in two different languages, they are equal to each other up to an additive constant.
(3)$$\left|K_{U}\left(x\right)-K_{V}\left(x\right)\right|<C_{UV}$$
where $K_{U}\left(x\right)$ (respectively, $K_{V}\left(x\right)$) is the Kolmogorov complexity with a universal machine *U* (respectively, *V*) and $C_{UV}$ is a constant.

This theorem allows us to state (with respect to a constant) that the change in the universal Turing machine (i.e., programming language) does not affect the Kolmogorov complexity of the sequence *x*. In this context, the shortest program can be considered the minimal description of the object and a satisfying approximation of the Kolmogorov complexity.

This algorithmic method enables the description to be seen as a compressed version of the object. The program computes the object with a fewer number of symbols than the number of symbols used to compose the object (Kφ(x)⩽|x|—incompressibility notion [10]).

In fact, the description of the object (the program) and the compressed version of an object are based on the same principle, i.e., the search for regularities. In an effort to find the shortest program or reduce the size of a file as much as possible, the common goal is to exploit the repeated information contained in the object.

The compression ratio (C(o)) is then an approximation of the Kolmogorov complexity since it supplies an indication of the size of an object’s description and thus its complexity.
(4)$$C\left(o\right)=\frac{s(o)}{s(K(o))}$$
where s(o) is the file size of the original object and s(K(o)) is the compressed file size of the original object.

In order to better understand this concept, let us come back to the two strings, $S_{1}$ = “cqzabrdiok” and $S_{2}$ = “aaabcddddd”, we used previously. These two objects have intuitively different complexities, which are, respectively, high and low (in the sense of Kolmogorov).

To determine the string complexity, the Run-Length Encoding (RLE) [16] simple lossless (lossless compression is a method in which the original data can be recovered exactly from the compressed data) data compression algorithm can be used (this compressor is, of course, far from being optimal in terms of compression but is used for illustration purposes). After RLE compression, the output of the first string is “1c1q1z1a1b1r1d1i1o1k” with a total of twenty characters and the output of the second string is “3a1b1c5d” with a total of eight characters. The outputs produced reflect the higher complexity of the first input string due to the absence of repetitions, whereas the second benefits from the repetitions. The Kolmogorov complexity is a measure of the complexity of a single object; it is not possible to know the complexity of several images simultaneously.

If we consider complexity as a measure of information, we can make the assumption that comparing the complexities of the objects can allow for the classification of the information represented by them.

#### 2.1.3. The Similarity between Two Objects

In the sense of Kolmogorov, two objects are similar in complexity if one can be described by the description of the other [1].

The normalized information distance (NID) [17] is thus a theoretical universal distance based on the Kolmogorov complexity and is defined as:$$\frac{\max\{K(x|y),K(y|x)\}}{\max\{K(x),K(y)\}}$$
where K(x|y) (respectively, K(y|x)) is the conditional Kolmogorov (i.e., the length of the shortest program to compute *x* if *y* is furnished as an auxiliary input to the computation [17]) of *x* (respectively, *y*) with respect to *y* (respectively, *x*).

Unfortunately, it has been proven to be uncomputable since *K* is uncomputable [18], but hopefully, it is approximable [1,19].

Cilibrasi and Vitanyi proposed a computable approximation of the NID referred to as the normalized compression distance (NCD) [1] using the idea presented above that considers compressors as an approximation of the Kolmogorov complexity:
(5)$$NCD\left(x,y\right)=\frac{C(xy)-min\{C(x),C(y)\}}{max\{C(x),C(y)\}}$$
where C(x) and C(y) denote, respectively, the compressed ratio of the object *x* and the object *y*. C(xy) represents the complexity of the concatenation (i.e., the operation joining two objects together) of object *x* and object *y*. The maximal similarity between objects is then represented by 0 and the maximal dissimilarity by 1+ϵ (ϵ is due to imperfections in the compression techniques) [1].

The compressor must be *normal* to compute this distance so it must satisfy the properties of idempotence, symmetry, monotonicity, and distributivity [1]. Indeed, a normal compressor detects repetitions in the data to be compressed, which enables the removal of redundant data in the description of the object. This notion allows NCD to approximate optimality and thus be a quasi-universal metric of similarity.

Idempotency Property
(6)C(xx)=C(x) Symmetry Property
(7)C(xy)=C(yx) Monotonicity Property
(8)C(xy)⩾C(x) Distributivity Property
(9)C(xy)+C(z)⩽C(xz)+C(yz)

Then, NCD has been proven to be a distance and thus respects the following properties:

Separation Property
(10)NCD(x,y)=0⇔x=y Symmetry Property
(11)NCD(x,y)=NCD(y,x) Triangle Inequality Property
(12)NCD(x,y)⩽NCD(x,z)+NCD(z,y)

The objective of Cilibrasi and Vitanyi was not to order objects but to group similar objects. The degree of similarity between two objects is important but insufficient for ranking information according to its complexity. Therefore, we decided to propose an alternative measure of information complexity. This measure will allow us to better isolate and analyze the impact of information complexity.

## 3. Information Complexity Ranking (ICR)

The main contribution of this article is the proposal of a new measure that allows for the ordering of several objects from the least to the most complex and thus facilitates their comparison. The fundamental idea of the ICR method is to provide a finer measure of complexity. From a purely algorithmic point of view, two images can be very similar, which NCD faithfully retranscribes, but cannot be compared in terms of complexity by NCD. It is this gap that we are attempting to fill by using the intrinsic complexity of each image.

Our purpose is to propose a partial order, i.e., a binary relation that respects the following properties: reflexivity, antisymmetry, and transitivity. Having a partial order allows us to coherently compare the elements to establish a ranking. This is not the case with NCD, which aims at grouping similar elements.

To provide these properties, we first introduce the concept of Information Complexity Ranking (ICR), which is based on measuring the similarity between two objects (i.e., NCD) and weighting it by the intrinsic complexity of each object (compression ratio) (C(x)). To ensure that we have a partial order, we incorporate the complexities of both pieces of information by multiplying NCD by the product of the two compression ratios.
(13)$$ICR\left(x,y\right)=\frac{NCD(x,y)\times C(y)}{C(x)}$$

We then claim that the binary relation R(x,y), which is defined by
(14)R(x,y)=xRy⇔ICR(x,y)⩾ICR(y,x)
is a partial order.

### 3.1. Proof

The objects treated here are generic and can be any sequence of binary symbols that form a bitset. As mentioned above, a partial order is a binary relation satisfying the following properties:(15)$$\left\{\begin{array}{cc} R(x,x) & Reflexivity\\ R(x,y)\;AND\;R(y,z)\Rightarrow R(x,z) & Transitivity\\ R(x,y)\;AND\;R(y,x)\Rightarrow x=y & Anti-symmetry \end{array}\right.$$

#### 3.1.1. Reflexivity Property

Due to the separation property (Equation (Equation 10)) of NCD, NCD(x,x) = 0 and, therefore, ICR(x,x)⩾ICR(x,x) and **R(x,y) is reflexive**.

#### 3.1.2. Transitivity Property

Let us suppose that ICR(x,y)⩾ICR(y,x) and ICR(y,z)⩾ICR(z,y).

We can rewrite it as
(16)$$\left\{\begin{array}{c} \frac{NCD(x,y)\times C(y)}{C(x)}⩾\frac{NCD(y,x)\times C(x)}{C(y)}\\ \frac{NCD(y,z)\times C(z)}{C(y)}⩾\frac{NCD(z,y)\times C(y)}{C(z)} \end{array}\right.$$

By multiplying both sides, respectively, by C(x)C(y) and C(y)C(z), (C(u) (being a positive function by definition), we obtain
(17)$$\left\{\begin{array}{c} NCD\left(x,y\right)\times C^{2}\left(y\right)⩾NCD\left(y,x\right)\times C^{2}\left(x\right)\\ NCD\left(y,z\right)\times C^{2}\left(z\right)⩾NCD\left(z,y\right)\times C^{2}\left(y\right) \end{array}\right.$$

Due to the symmetry property of NCD (Equation (Equation 11)) and as NCD is positive, we can divide both sides, respectively, by NCD(x,y) and NCD(z,y). We then obtain
(18)$$\left\{\begin{array}{c} C^{2}\left(y\right)⩾C^{2}\left(x\right)\\ C^{2}\left(z\right)⩾C^{2}\left(y\right) \end{array}\right.$$

From these equations, we can deduce that $C^{2}\left(z\right)⩾C^{2}\left(x\right)$.

By multiplying both sides by NCD(x,z) and due to the symmetry of NCD, we have $NCD\left(x,z\right)\times C^{2}\left(z\right)⩾NCD\left(z,x\right)\times C^{2}\left(x\right)$. Dividing both sides by C(z)×C(x) allows us to conclude if C(x)⩾0 and C(z)⩾0 so that
$$\frac{NCD(x,z)\times C(z)}{C(x)}⩾\frac{NCD(z,x)\times C(x)}{C(z)}$$
can be rewritten as
ICR(x,z)⩾ICR(z,x)
which proves the transitivity of the relation.

#### 3.1.3. Anti-Symmetry Property

Let us suppose that ICR(x,y)⩾ICR(y,x) and ICR(y,x)⩾ICR(x,y).

We thus have
(19)$$\left\{\begin{array}{c} \frac{NCD(x,y)\times C(y)}{C(x)}⩾\frac{NCD(y,x)\times C(x)}{C(y)}\\ \frac{NCD(y,x)\times C(x)}{C(y)}⩾\frac{NCD(x,y)\times C(y)}{C(x)} \end{array}\right.$$

As NCD is symmetric, we can divide both sides by NCD(x,y), which gives
(20)$$\left\{\begin{array}{c} \frac{C(y)}{C(x)}⩾\frac{C(x)}{C(y)}\\ \frac{C(x)}{C(y)}⩾\frac{C(y)}{C(x)} \end{array}\right.$$

Multiplying both sides by C(x)×C(y) gives
(21)$$\left\{\begin{array}{c} C^{2}\left(y\right)⩾C^{2}\left(x\right)\\ C^{2}\left(x\right)⩾C^{2}\left(y\right) \end{array}\right.$$
which can only be true if C(x)=C(y), allowing us to conclude that the relation is anti-symmetric.

To conclude, as R(x,y) is reflexive, transitive, and anti-symmetric, it is a partial order.

Our method is an abstract measure that relies on strong hypotheses concerning the properties of both the compressor and the concatenation operator and must be validated in concrete applications. First, we validate our hypotheses on a benchmark experiment and then, we apply our method to a reference database in the domain of complexity measurement.

## 4. Experiments

### 4.1. Synthetic Dataset

#### 4.1.1. Purpose of the Experiment

The Kolmogorov complexity, the central theoretical notion of our ICR proposal, is not computable but is approximable. This approximation, which is also useful for NCD, is performed with the help of compression algorithms. These compression algorithms must be normal. With this aim in mind, we wanted to test a panel of eight algorithms from the literature to determine whether they respect the properties of a normal compressor and thus use them for our image classification method.

#### 4.1.2. Protocol

To compute the complexity of an image, the original image format must use a lossless format (no compression). The bitmap (BMP) format respects this prerequisite and can be edited, manipulated, and translated without losing image quality.

The images used for testing were generated randomly by assembling three kinds of pictorial elements that can be easily synthesized: text, digits, and colored dots. The resulting image consisted of a string of 20 characters concatenated with a number containing 1 to 4 digits and colored dots whose colors were randomly selected. This template was not entirely random for two reasons: first, we wanted to compare similar structures with slight differences to test the sensitivity of our method, and second, this format was consistent with the visual labels we intend to use in a future experiment to examine the impact of information complexity on the decision process (such as nutrition labels [20]).

One of the hypotheses for our method to be consistent is that the concatenation operation should respect certain properties. As Kolmogorov takes the order of information into account, it was important to check the influence of the direction of concatenation for each image pair. To this end, the images were concatenated in pairs, either vertically or horizontally.

Another important hypothesis concerning the applicability of our method concerned the quality of the compressor. To determine whether the compression algorithms could influence the results, we tested several algorithms for computing the Kolmogorov complexity. In practice, eight commonly used lossless compression algorithms were tested (Figure 1 and Figure 2).

The parameters of these compressors were also studied and the parameter maximizing the compression ratio was chosen.

For the concatenation of the images, we resized the largest image by taking the vertical dimension of the smallest image to avoid introducing artifacts by stretching the smallest image. Overall, 7200 images were created (3600 images were horizontally concatenated and 3600 images were vertically concatenated).

For the compression algorithms, we tested how much they respected the properties required for NCD in an effort to ensure our method was generalizable (Table 1), as well as the resulting compression ratios.

#### 4.1.3. Results

The objective of our comparison was to study how the different compressors behaved regarding the expected properties of a normal compressor, i.e., idempotency, symmetry, monotonicity, and distributivity. Rather than using a boolean value for each of the four properties, we wanted to determine the loss generated by compressing two identical images (detection of repetition) or whether a compressor can detect the concatenation order of two different images. For instance, if a compressor was idempotent (C(xx)=C(x)), then it should compress similarly an image *x* and a duplication of this image xx. However, as we can see in the first line of Table 1, no compressor achieved a value of 0, even though PNG was less affected than the other compressors. We can also see that Huffman was not sensitive to symmetries (Table 1, lines 3 and 4, last column) since it only used the frequency of each symbol and not the relative positions. Therefore, using these values rather than a boolean value allowed us to compare compressors more accurately.

The PNG, PPM, and LZ77 algorithms achieved the best results regarding the idempotency property (C(xx)=C(x)), which allowed the compressor to detect exact repetitions and thus achieve reasonable accuracy. With regard to the symmetry property (C(xy)=C(yx)), which is important for the optimal use of NCD, the PNG and LZ77 algorithms presented the best results in horizontal and vertical concatenation. The PPM algorithm, as part of the family of predictive compressors, showed small deviations, which will disappear asymptotically with the length of x and y [1]. Concerning the monotonicity property (C(xy)⩾C(x)), which allows for the preservation of the concatenation order of the information, an important notion for NCD, the three selected algorithms presented similar results to the other tested algorithms, except for Huffman. Similar results were found for the notion of distributivity (C(xy)+C(z)⩽C(xz)+C(yz)). Regarding the compression ratio performance, the PNG, LZ77, and PPM algorithms achieved similar results to the best-performing compressor, the LZW.

Note that the results were obtained using the maximum compression parameters of each algorithm.

In conclusion, in order to maximize the accuracy of NCD and, consequently, our ICR proposal, based on the results presented in Table 1, three compressors, PNG, PPM, and LZ77, were chosen for further experimentation on the SAVOIAS dataset.

### 4.2. SAVOIAS Dataset

#### 4.2.1. Dataset Description

To further test the performance of our approach in a more realistic context, we tested its robustness on the SAVOIAS dataset [21]. The Boston University team that provided this dataset intended to create a visual complexity dataset of more than **1400 images**. One of its advantages is the relevance of the choice of images to address the problem of visual complexity. The diversity of the images is also interesting with various low-level and high-level features. The images are clustered into seven categories, namely Scenes, Advertisements, Visualization and infographics, Objects, Interior design, Art, and Suprematism. They compared their ground truths to five state-of-the-art complexity algorithms to measure their performance. We used these five algorithms for comparison with our method. The ground truths of the dataset were obtained by asking **1600 contributors** to compare a subset of pairs of images in a set of more than **37,000 pairs of images**. Participants were asked to estimate the visual complexity of each pair by giving a score on a scale from 0 to 100, with 0 being very simple and 100 very complex [21].

To illustrate, let us use as an example two images found in the dataset (Figure 3). The output of ICR for these two images is then ICR(A,B) = **4032** and ICR(B,A) = **0.229**. Thus, we rank image A as more complex than image B, which seems intuitively consistent.

To go beyond the first “intuitive” impression, we proposed a rigorous experimental protocol.

#### 4.2.2. Protocol

We used the ground truths of the SAVOIAS experiment to compare the state-of-the-art visual complexity algorithms and the information complexity ranking (ICR). Table 2 presents the details of the number of images by category and the number of resulting comparisons.

At the end of the process, we obtained four values for the ICR computation. The first two values corresponded to the calculation of ICR(A,B) for which image A was concatenated with image B in the horizontal (respectively, vertical) direction, whereas the last two values corresponded to ICR(B,A), the concatenation of the swapped images. To compare the values obtained and the ground truths, we ranked the images by complexity. For this purpose, we use Condorcet’s method [22]. This voting system, which was established in 1785 by Nicolas de Condorcet, allows for the designation of the winner of an election by contrasting the number of votes in a head-to-head comparison. In the application of this method, the maximum value of each comparison by pair was considered a “vote”. All the “votes” were then summed for each image to obtain a ranking of all the images. Finally, in order to compare the rank of complexity given by the ground truths and the rank given by our method, we computed the Spearman correlation.

To demonstrate the robustness of our proposal, we also compared it to five other state-of-the-art visual complexity measures, namely the edge density, feature crowding, subband entropy [23], number of regions [24], and compression ratio [25] measures. The edge density measure is the percentage of pixels that are edge pixels in an image. Feature clutter represents visual clutter and the difficulty of adding a new element that can reliably draw attention to a display or scene. Subband entropy is based on the idea that clutter is related to the number of bits required to encode the image into subbands (wavelets). Measuring the number of regions is equivalent to counting the densest regions in the feature space of an image. The compression ratio corresponds to the bppR (bit-per-pixel ratio), which is the ratio of the bpp of a distorted image to the bpp of a lossless compressed version of the original image (in JPEG). For generalization purposes, it is important to remember that ICR can be applied to all types of data, unlike these methods that are only applicable to images.

#### 4.2.3. Results

Our proposal is based on a combination of NCD and the compression ratio (ICR(x,y)=NCD(x,y)×C(y)/C(x)) so it is important to demonstrate its performance with respect to the individual elements that comprise it. As shown in Table 3, our ICR method had better correlations with human perception in all dataset categories compared to using NCD alone.

By comparing the results in Table 3 and the results of the compression ratio in Table 4, we can see the contribution of NCD in the categories *Scenes, Advertisement, Art, and Suprematism*. In the categories *Objects* and *Interior decoration*, the results of our method were similar to those of the compression ratio alone and superior to those of the NCD alone. The coefficient of the *Visualizations* category was less efficient because the images in this category were very heterogeneous so the contribution of NCD was almost non-existent.

As seen in Table 4, our method achieved the best results for the category *Advertisement* and *Interior Design*. The results of our method for the categories *Art* and *Suprematism* were better than those of the edge density, feature congestion, subband entropy, and compression ratio measures. In the *Visualizations* category, the results of our method were lower than expected compared to those of the other methods. Concerning the images representing objects, the performance of our method was below that of the other measures except for the subband entropy measure.

The difference for the category *Visualizations* (Figure 4) can be explained by the large diversity of the images, which involved a low similarity. As this notion is central to our proposal, the results produced were worse than in the other categories. The Suprematism painting style, for its part, is based on geometric forms such as circles, squares, or rectangles. It is also characterized by a reduced color palette. It reinforces the capability of our measurement to better rank the complexity by detecting the numerous repetitions contained in these images. The possible improvements are developed in the next section. Note that whatever the direction of concatenation (horizontal or vertical), the correlation coefficient of our approach was identical, which demonstrates that the direction of concatenation did not affect the complexity of the image.

## 5. Conclusions and Perspectives

Quantifying information complexity is an important area of research in adaptive systems, decision support, and consumer behavior. To improve this understanding, we propose a new method for prioritizing information according to its complexity. This ranking is accompanied by a new proposal for comparing objects. It is based on the similarity between two objects (normalized compression distance) and the intrinsic complexity in the sense of Kolmogorov. The weighting of the similarity of several objects by their intrinsic complexity allows us to mark up the most complex object, and thus rank the objects to evaluate their potential involvement in decision-making processes.

To validate our approach, we tested it on *the SAVOIAS* dataset [21]. To stay within the scope of complexity as a measure of information, we compared it to the compression ratio measure used in the dataset. The results of our proposed method were comparable to those of the compression ratio for most of the image categories. For images that contained less context or fewer semantics such as works of art, our metric performed better than the compression ratio measure. For images with text or graphics, our approach achieved lower performance.

This can be explained by taking into account that the semantic aspect of the information is rather weak in our approach. Our goal was to build a measure of information in order to understand its influence and the semantic aspects of the information will be part of our future research. The integration of a more user-centered perspective (emotional aspect) is also planned to gain a deeper understanding of people’s behavior toward the information that is transmitted to them.

In our future experimental works, we plan to conduct a new experiment using eye-tracking methods to demonstrate a possible correlation between our ICR ranking method and the perception of complexity in humans. In future studies, we will mobilize different notions, such as the decision process or the role of visual attention on choice, in order to quantify a level of informational complexity that induces a particular choice or response from individuals.

Given the promising results on images, it would also be interesting to test ICR in other use cases, such as Natural Language Processing (NLP), cybersecurity with malware detection [26], or even in genetics and phylogeny, based on the initial experiments presented using the NCD definition [1].

In our future theoretical works, we would also like to improve our approach centered on the complexity of the object by using the logical depth of Bennett [27]. The Kolmogorov complexity is considered a good estimator of an object’s complexity that is represented by a binary sequence. It is a useful notion for defining the absolute notion of a random sequence [28] but is not able to capture the notion of structured complexity [29]. Bennett tried to measure the real value of the information contained in the sequence, with the contribution of computational content. This would improve the accuracy of our ICR ranking method.

## Figures and Tables

**Figure 1 entropy-25-00439-f001:**
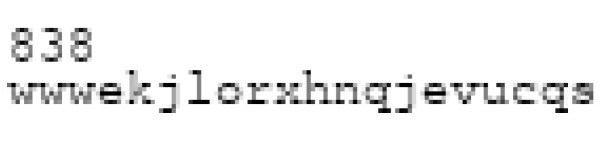
Example of vertical concatenation of a generated image.

**Figure 2 entropy-25-00439-f002:**
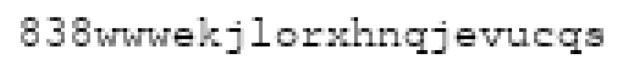
Example of horizontal concatenation of a generated image.

**Figure 3 entropy-25-00439-f003:**
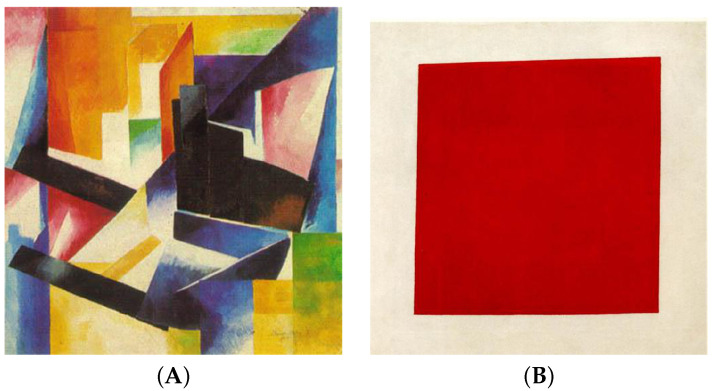
Example of two images (left, IMAGE (**A**), and right, IMAGE (**B**)) used to test our method of Information Complexity Ranking (ICR).

**Figure 4 entropy-25-00439-f004:**
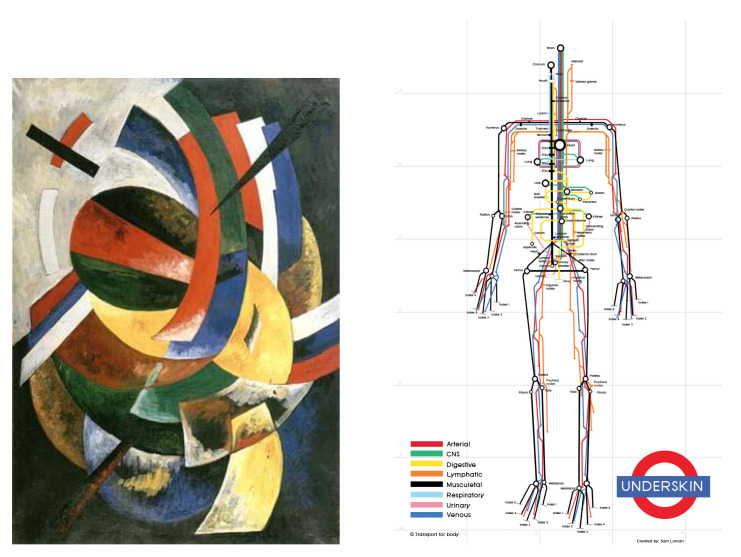
Examples of *Suprematism* and *Visualizations* categories.

**Table 1 entropy-25-00439-t001:** Performance of compression algorithms by calculating the compression ratio for each property required to satisfy the concept of a normal compressor.

Algorithm	PNG	PPM	LZ77	ALZ	LZP	LZW	AC	Huffman
Idempotency horizontal	0.088	0.091	0.098	0.196	0.209	0.555	0.533	0.216
Idempotency vertical	0.043	0.047	0.058	0.130	0.133	0.631	0.591	0.237
Symmetry horizontal	0.004	0.014	0.004	0.007	0.007	0.004	0.001	0.000
Symmetry vertical	0.006	0.027	0.009	0.031	0.005	0.007	0.009	0.000
Monotonicity horizontal	0.318	0.202	0.285	0.197	0.334	0.323	0.731	1.423
Monotonicity vertical	0.344	0.204	0.298	0.199	0.329	0.343	0.909	1.588
Distributivity horizontal	0.313	0.197	0.273	0.203	0.327	0.314	0.736	1.414
Distributivity vertical	0.329	0.202	0.287	0.203	0.313	0.333	0.894	1.549
Compression ratio	3.174	3.676	3.509	3.745	3.300	3.875	2.028	0.760

**Table 2 entropy-25-00439-t002:** Number of images generated to test ICR. Factor 2 applied to the *Number of comparison* columns represents the horizontal and vertical concatenation of each image.

Category	Number of	Number of	Total Number of
	Images	Comparisons	Processed Images
Scenes	200	(200×200)×2	80,000
Advertisement	200	(200×200)×2	80,000
Visualization	200	(200×200)×2	80,000
Objects	200	(200×200)×2	80,000
Interior Design	100	(100×100)×2	20,000
Art	420	(420×420)×2	352,800
Suprematism	100	(100×100)×2	20,000

**Table 3 entropy-25-00439-t003:** Comparison of ICR and NCD. The NCD and ICR values are the averages of the PNG, PPM, and LZ77 algorithms. The best result for each category is highlighted in bold.

Category	NCD	ICR
Scenes	0.23	**0.40**
Advertisement	0.16	**0.58**
Visualizations	0.13	**0.31**
Objects	0.05	**0.16**
Interior Design	0.52	**0.67**
Art	0.28	**0.58**
Suprematism	0.47	**0.73**

**Table 4 entropy-25-00439-t004:** Comparison of ICR and visual complexity measurements in the literature. The values of the first five columns have been carried over from the corresponding columns of Table 3 in [21] (Adapted from [21]). The best result for each category is highlighted in bold.

Category	Edge Density	Number of Regions	Feature Congestion	Subband Entropy	Compression Ratio	ICR
Scenes	0.16	**0.57**	0.42	0.16	0.30	0.40
Advertisement	0.54	0.41	0.56	0.54	0.56	**0.58**
Visualizations	0.57	0.38	0.52	**0.61**	0.55	0.31
Objects	0.28	0.29	**0.30**	0.10	0.16	0.16
Interior Design	0.61	0.67	0.58	0.31	**0.68**	0.67
Art	0.48	**0.65**	0.22	0.33	0.51	0.58
Suprematism	0.18	**0.84**	0.48	0.39	0.60	0.73

## Data Availability

Not applicable.

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
