# Peer review of "Information Complexity Ranking: A New Method of Ranking Images by Algorithmic Complexity"

_entropy, 2023, doi:10.3390/e25030439_

Round 1

Reviewer 1 Report (Previous Reviewer 3)

I appreciate the premise of defining a complexity metric to evaluate image complexity based on compression ratio and NCD. The SAVOIAS dataset is used to demonstrate the applicability of the metric for correlation with the ground truth (human estimates of visual complexity).

- MAJOR: I am concerned about the missing motivation (in the introduction section) behind introducing the new ICR method (line 52), i.e. some pain point in applications that currently available methods like NCD cannot address and needs ICR.

- MAJOR: The phrases containing "behavioral/behavior" (e.g. lines 22, 24, 111, 120, 222, 420, 421) or "decision" (e.g. lines 186, 227, 287, 421, 427) stress too much on the impact of this article on human decision and behavioral pattern. The article at this stage is focused on the definition of the ICR and is yet to demonstrate that, and is left for future work. The article's title and Line 49: "to measure the complexity of each image and compare them to establish a ranking" reflect the aim of this research more humbly and I suggest not highlighting ICR's aid in human behavior and decision-making till further study.

- MINOR: It is not clear what led to that specific definition of ICR in equation 13 and the importance of partial order. 

- MINOR: The section of Shannon entropy can be significantly reduced since it is not part of the ICR.

I appreciate that the authors are interested in including Logical depth in their future work as it is known to have a higher correlation to perceived complexity by human cognition.

Author Response

Reviewer 2 Report (New Reviewer)

seems like a decent experimental paper overall to me. ICR is an interesting variation and i find it plausible that it does improve image recognition. i have two questions:

1. would ICR work better in other non-image contexts as well? given the general structure of the modification to NCD i am not sure why it would be helpful specifically to images. maybe try more domains with experiments (like human language text or genomic info may be convenient). i noticed the formal presentation of ICR is quite general but then the experiments are all only in the image domain so would be curious about results outside of images.

2. can the theoretical foundation / justification for ICR be strengthened?

intuitively i can imagine how the compression ratio multiplication can help. but i would like to see more than just the proof that it induces a partial ordering if possible.

Round 2

Reviewer 1 Report (Previous Reviewer 3)

My concerns in the 1st phase of the review have been sufficiently addressed.

Reviewer 2 Report (New Reviewer)

thanks for adjustments

This manuscript is a resubmission of an earlier submission. The following is a list of the peer review reports and author responses from that submission.

Round 1

Reviewer 1 Report

Authors' information complexity ranking is ICR(x,y)=NCD(x,y) C(y)/C(x). However, we note that if it is defined as  ICR2(x,y)=C(y)/C(x), the results (properties) are almost the same. Moreover, the results (properties) just from C(y)>C(x) or C(y)<C(x). This ranking just is C(y)>C(x) or C(y)<C(x).

Line 224-227,why these sentences are in bold.

Reviewer 2 Report

This paper proposes a new method for ranking the information content of
digital objects (ICR, Information Complexity Ranking), and compares it
to the empirical ranking derived from an existing database of human
judgements of image perceptual complexity.

The proposed method is a minor modification of the Normalized
Compression Distance. While the particular quantity proposed in the
paper may indeed be novel, the paper is severely lacking on multiple
fronts. In its present form, it doesn't seem salvageable, and I
recommend rejection. I will outline below some of my concerns.

1. The paper is very hard to read. Language-wise it's not great
   (although this difficulty is not insurmountable for someone who is
   familiar with the particular writing accent possessed by the
   authors), but more importantly several passages simply do not provide
   enough information to convey their intended message. For instance, it
   is almost impossible to understand what section 3.1, and in
   particular 3.1.2, is talking about. In this section, various
   compression algorithms are "tested" against some "properties" such as
   idempotency, symmetry etc. As far as I could tell, these properties
   are stated as being necessary for NCD, but they are never
   defined. What does it mean for a compressor to possess symmetry (or
   idempotency etc)? Moreover, whatever the definition of any of these
   property may be, what does it mean to "test" a compressor against
   them? Naively, one would expect that a certain compressor either has
   or does not have a certain property. Why is the outcome of such a
   test a continuous number (a compression ratio) rather than a logical
   value (true/false)? And further, why is the smallest compression
   ratio highlighted as the best in Table 1, while the largest
   compression ratios are highlighted as the best in Tables 2 and 3?
2. There is a wide, unjustified logical disconnect in the premise of the
   paper. While the core content of this work is the definition and
   examination of ICR, the main use case put forward for ICR is for the
   analysis of the complexity of information provided to human subjects
   in decision-making contexts. More specifically, the ultimate purpose
   of this research seems to be to identify "the ideal level of
   complexity of information to change behavior". But the link between
   "information complexity" (a term that is very loosely used throughout
   the paper, and never defined - what is the difference between
   information and information complexity?) and behavior is simply
   assumed: it is stated that some "ideal level" of complexity exists at
   which it may be possible to more efficiently manipulate people's
   decision making process, but this crucial point is never justified or
   supported by any evidence. The stated goal of the project hinges on
   this single point of failure, so it is very puzzling to see it
   completely unaddressed.
3. Apart from the criticisms above, the proposed measure (which, again,
   is just a minor modification of NCD) seems to perform systematically
   worse than other, existing metrics in the only benchmark provided
   that is relevant to the stated goal of the research (section 3.2,
   where it is asked whether ICR captures humans' intuitive notion of
   image complexity). From Table 5, we can see that the "number of
   regions" approach outperforms ICR in 6 out of 7 image
   categories. Therefore, the contribution of this work can't be
   salvaged on purely pragmatic grounds (even if one was willing to
   overlook the major points discussed above).
4. The paper is peppered with puzzling, confusing, incomplete or
   outright incorrect statements. For instance:
   • line 28-29: "therefore, what could be relevant for the decision
     should be the relative complexity". This is a non-sequitur; I
     honestly have no idea how it could be justified by the preceding
     sentence.
   • section 2.1 contains multiple mistakes and confusing passages. For
     instance, the fact that "the less frequent is a message, the more
     effort is needed to transmit it" is presented as a consequence of
     the notion of entropy, but this makes no sense. One can of course
     connect entropy and information theory with coding theory (as the
     authors do when they mention Huffman coding), but there is no way
     in which the "effort" needed to transmit a message is necessarily
     higher for infrequent messages. For instance, one can imagine a
     code that encodes infrequent messages in shorter codes and frequent
     messages in longer ones; this code is perfectly possible and can't
     be used, it's just suboptimal.
   • lines 136-137: "this theorem [Kolmogorov's invariance] let us
     expect that the length of any program that can generate an
     information is an approximation (with respect to a constant) of the
     Kolmogorov complexity". This is false, and suggests a worrying lack
     of understanding of the concepts being discussed here. Kolmogorov
     invariance means that, if you have an object x and you consider the
     shortest program expressing x in two programming languages, the
     difference between the length of these two programs will be bounded
     by a constant that depends only on the choice of programming
     languages. In other words, it means that a choice of programming
     language doesn't matter in the definition of Kolmogorov
     complexity. But this is very different from what is asserted in the
     passage I just cited: the specific program being used matters! For
     instance, for a given object x and a choice of programming
     language, I can generate an arbitrarily long program by summing and
     then subtracting π to some numerical variable, where π is specified
     in the source code of the program up to a sufficiently large number
     of decimal digits. Therefore, the authors' statement here (that the
     length of any program is an approximation of Kolmogorov complexity,
     up to a constant) is not just demonstrably false, but shows a lack
     of understanding of the main theoretical notions being discussed.
   • The second paragraph of the introduction contains the statement "we
     intend to use a mobile application that allows us to control the
     main variable which is the information provided to the person, and
     thus validate or invalidate the hypothesis…". Having read this, I
     honestly expected to find the experiment in the paper (and I
     thought that "we hope to" was an odd choice of phrasing). It turns
     out that this experiment hasn't been performed yet, and it's
     presented as future research in the discussion! I found this
     extremely confusing - please keep future research within the
     discussion section only, or clearly signpost that you're talking
     about something you haven't done yet.

I will not list minor points, language suggestions and typos, as I
believe that the major points above are enough to prevent any
recommendation towards acceptance of this work (including with major
revisions).

Reviewer 3 Report

The motivation of the research - of using approximate AIT metrics within a practical use case is highly appreciated. To improve the quality of the article, I suggest the following changes:

Major:

  • While the title, introduction, and conclusion mentions 'decision making', it is unclear how the ICR poset will help in that. Please clarify that connection.
  • Line 192 is the core 'theoretical' motivation, and should be mentioned in the introduction.
  • Since the goal is to help in decision-making, it might be important to mention the connection to Artificial General Intelligence models like AIXI or QKSA.

Minor:

  • Lines 64-69 are easier to understand if presented as two Huffman trees.
  • Line 32 in the introduction mentions 'intend to use a mobile application'. It is not clear that it is future work (and only becomes clear in line 364).
  • The explanation of logical depth can be reduced slightly (lines 379 to 388 don't add too much value). It is a well-motivated future direction that can already be appreciated from lines 371 to 379)
  • Three additional notions that might be interesting to mention: Zenil's Block Decomposition Method; Baez's Algorithmic Thermodynamics; Schmidhuber's Speed-prior.